# Genetic risk for autoimmunity is associated with distinct changes in the human gut microbiome

Jordan T. Russell [1], Luiz F.W. Roesch [2], Malin Ördberg [3], Jorma Ilonen[4], Mark A. Atkinson [5,6], Desmond A. Schatz[6], Eric W. Triplett [1] & Johnny Ludvigsson[3]

Susceptibility to many human autoimmune diseases is under strong genetic control by class II human leukocyte antigen (HLA) allele combinations. These genes remain by far the greatest risk factors in the development of type 1 diabetes and celiac disease. Despite this, little is known about HLA influences on the composition of the human gut microbiome, a potential source of environmental influence on disease. Here, using a general population cohort from the All Babies in Southeast Sweden study, we report that genetic risk for developing type 1 diabetes autoimmunity is associated with distinct changes in the gut microbiome. Both the core microbiome and beta diversity differ with HLA risk group and genotype. In addition, protective HLA haplotypes are associated with bacterial genera *Intestinibacter* and *Romboutsia*. Thus, general population cohorts are valuable in identifying potential environmental triggers or protective factors for autoimmune diseases that may otherwise be masked by strong genetic control.

[1] Department of Microbiology and Cell Science, Institute of Food and Agricultural Sciences University of Florida, Gainesville, 32611-0700 FL, USA. [2] Biological Sciences, Universidade Federal do Pampa, São Gabriel 97300-000, Brazil. [3] Crown Princess Victoria's Children's Hospital, Region Östergötland, Division of Pediatrics, Linköping University, Linköping, SE 58185, Sweden. [4] Immunogenetics Laboratory, Institute of Biomedicine, University of Turku, and Clinical Microbiology, Turku University Hospital, Turku 20521, Finland. [5] Department of Pathology, University of Florida Diabetes Institute, Gainesville, 32610 FL, USA. [6] Department of Pediatrics, College of Medicine, University of Florida, Gainesville, 32610 FL, USA. Correspondence and requests for materials should be addressed to E.W.T. (email: ewt@ufl.edu)

Type 1 diabetes mellitus (T1D) is an autoimmune disorder characterized by the destruction of insulin-producing ß-cells in the pancreas, resulting in a life-long dependence on exogenous insulin. T1D development is thought to be driven by both genetic and environmental influences, as genetic susceptibility alone is not enough to cause disease[1]. Environmental factors are considered to be important in triggering the onset of disease development in genetically susceptible individuals; however, identifying universal triggers remains a challenge[1,2].

Genetic risk factors for T1D have been identified in over 50 diverse genetic loci but the greatest genetic determinant of T1D remains the human leukocyte antigen (HLA) region[3]. HLA genes are highly polymorphic, resulting in sequence variations that have been shown to be both detrimental (e.g., HLA-DR3, DQB1*0201 and HLA-DR4, DQB1*0302) and protective (e.g., HLA-DR2, DQB1*0602) with regard to T1D susceptibility[4]. These polymorphisms result in distinct changes to the class II major histocompatibility complex (MHC) at the amino acid level, thereby altering the structure and peptide-binding capabilities of the molecule and thus, the presentation of antigens[5]. HLA polymorphisms could therefore be a driving force in controlling immune responses to microbes in the gut, a proposed environmental risk factor of T1D development[6].

The effect that MHC class II variation has on gut microbiome composition has been explored previously in mice[6–8]. For example, Kubinak et al. were able to demonstrate that the inability to present class II antigens led to distinct changes in gut microbial composition and structure, namely a decrease in *Lactobacillus* species and an enrichment of segmented filamentous bacteria[8]. More pertinently, they showed that gut microbial communities of the congenic mice were distinct based on MHC II polymorphisms that control T follicular helper cell influence on antibody response to commensal bacteria and that the effect of these polymorphisms was most potent at the mucosal interface of the gut vs. the feces[8]. Furthermore, results obtained by Silverman et al. showed that protection against insulitis in NOD mice transgenic for the missing MHC II Eα gene promoter led to distinct changes in the gut microbial community, and the protective effect was reliant on both the genotype and the gut community but not exclusively one or the other[6]. It is therefore feasible that MHC II genotype is important in not only shaping the bacterial community in the gut, but also in how the genetic and environmental components interact within the host leading to progression of or protection from autoimmune disease.

The intestinal microbiome has attracted a great deal of interest as a potential environmental component of T1D, as it is known to interact with and influence the immune system of the host[9–11]. Many previous and ongoing studies have aimed to identify organisms within the gut that may play a role in either the development or prevention of T1D[12–15]. Importantly, those efforts focused only on subjects at high genetic risk for T1D. In the Finnish Type 1 Diabetes Prediction and Prevention (DIPP) study, a high abundance of *Bacteroides dorei* preceded T1D autoimmunity by several months[12]. In the German BABYDIET study, no associations were found between anti-islet autoimmunity and bacterial diversity or composition[16], while analysis of bacterial co-occurrence networks revealed an association with butyrate production in controls[17]. In the multinational Environmental Determinants of Diabetes in the Young (TEDDY) study, only a few weak bacterial associations with T1D autoimmunity were identified[18,19].

Given that mostly weak bacterial associations with T1D autoimmunity have been observed in high T1D genetic risk cohorts, the question arose whether genetic risk alone imparts a dysbiosis of the microbiome and thus, whether any effect HLA may have on the microbiome may be masked in a high genetic risk cohort.

Here, a microbiome analysis of stool samples from the ABIS general population cohort shows the effect of HLA alleles on the human gut microbiome composition. Bacterial taxa negatively and positively associated with genetic risk for type 1 diabetes are identified.

## Results

**Description of cohort and study design**. The ABIS cohort has enrolled 17,055 newborn babies from Southeast Sweden born between 1 October 1997 and 1 October 1999. All mothers of babies born during this time period were invited to participate. This cohort serves as a large biobank of biological specimens obtained longitudinally from the enrolled children at birth, 1 year, 2–3 years, and 5–6 years of age. Collected samples types include blood, urine, stool and hair. In addition, parents of enrolled children completed questionnaires including information on duration of breastfeeding, antibiotic use, diet, etc. Many of the children enrolled also had their HLA genotype determined. The aim of the ABIS cohort, in part, is to identify the importance of environmental factors in autoimmune diseases (e.g. type 1 diabetes) and how genetic and environmental factors may interact in such diseases.

In the present study, we used high-throughput 16S rRNA sequencing to assess the microbiome of stool collected at 1 year of age from ABIS children. This time point was chosen due to the proximity in timing for development of T1D autoimmunity. Because HLA genotype data was available for only some of the children at the time of analysis, 403 individual 1-year stool samples were used. Associations between the 16S data and HLA genotype information for these children were made using multiple common statistical methods. Additionally, culturing methods were used to asses stool bacterial viability.

**HLA genetic risk explains microbiome variation**. Rarefaction curves for the observed number of amplicon sequencing variants (ASVs) and the Shannon alpha diversity index show that the chosen depth of 10,000 reads per sample was sufficient to represent the diversity of unique sequences in each sample (Fig. 1a, b). Shannon diversity was not significantly different overall among the genetic risk groups (Kruskal–Wallis, $p$-value = 0.4906), nor through pairwise comparisons between groups (Fig. 1c). The lack of difference in diversity between risk groups suggests that HLA does not have an effect on gut bacterial diversity. Antibiotic use (yes/no) within the first year (PERMANOVA: $R^2 = 0.00522$; F Model = 1.0526; $p$-value = 0.285), duration of exclusive breastfeeding (PERMANOVA: df = 10; $R^2 = 0.0241$; F Model = 0.97047; $p$-value = 0.6), mode of delivery (PERMANOVA: $R^2 = 0.00504$; F Model = 1.015, $p$-value = 0.395) and gender (PERMANOVA: $R^2 = 0.00242$; F Model = 0.97412; $p$-value = 0.529) did not have significant effects on gut bacterial composition at 1 year of age. Also, antibiotic use within 1 month prior to sample collection ($n = 30$) did not significantly impact gut bacterial composition (PERMANOVA: $R^2 = 0.00282$; F.Model = 1.1386; $p$-value = 0.168). Out of the total 403 subjects analyzed, 43 were still at least partially breastfed at the time of the 1-year sample collection. Gut microbiome composition was significantly impacted by breastfeeding status at the time of stool sample collection (PERMANOVA: $R^2 = 0.00966$; F.Model = 3.92; $p$-value = 0.001). Because breastfeeding status at the time of sample collection could be a confounding variable, this was corrected or accounted for in all subsequent analyses.

HLA risk for T1D was able to explain significant differences in gut microbiome composition (PERMANOVA: $R^2 = 0.0097$; F Model = 1.3088, $p$-value = 0.01, distance metric: binomial). Pairwise comparisons between each group showed that all but the

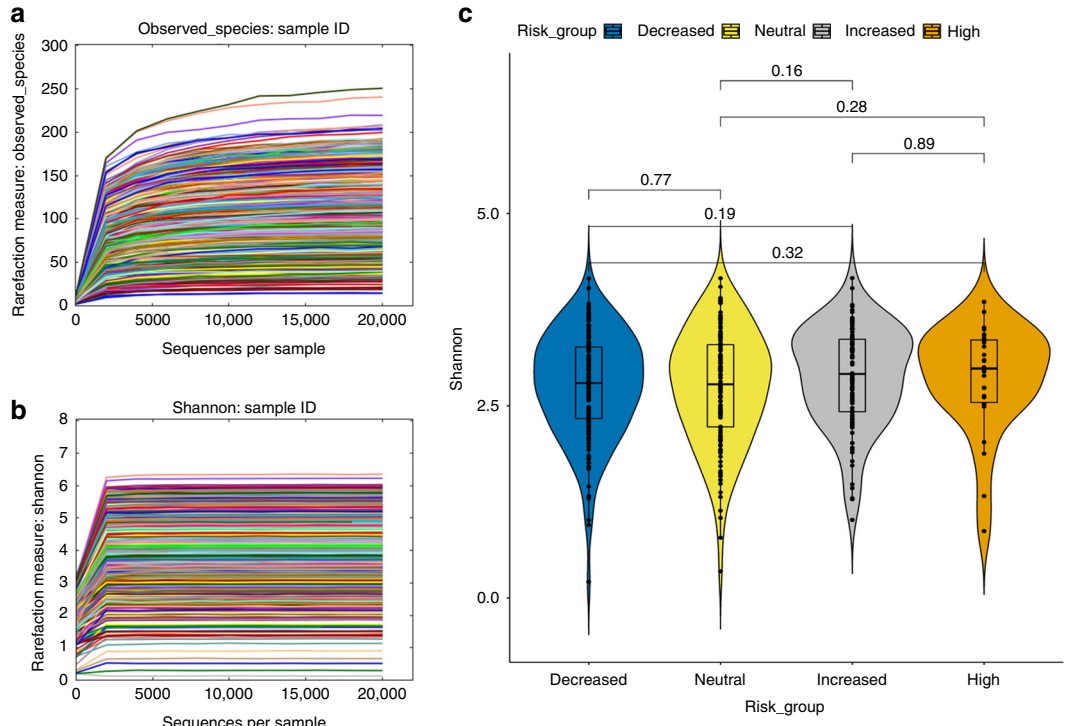

**Fig. 1** No significant difference in microbial alpha diversity between genetic risk groups. **a, b** Rarefaction curves were generated based on both the number of unique amplicon sequencing variants (**a**) and the Shannon diversity index measure (**b**) for each sample. **c** Violin box plots and pairwise statistical comparison (Wilcoxon test) between alpha diversities among genetic risk groups. *P*-values for each comparison are depicted above the boxplots of the groups being compared. For statistical testing, $n = 403$ independent stool samples. Boxplot medians (center lines); interquartile ranges (box ranges); whisker ranges: Decreased = 2.795; 2.335–3.262; 0.210–4.155, Neutral = 2.798; 2.226–3.299; 0.345–4.156, Increased = 2.915; 2.425–3.362; 1.015–4.160, High = 2.983; 2.545–3.353; 0.871–3.851. Source data for Fig. 1c are provided in the source data file

high vs. increased and decreased vs. neutral group comparisons were significantly different after correction for false discovery rate (FDR) (Table 1). Note that both high and increased risk groups are comprised of DR3 and/or DR4 haplotypes without protective haplotypes. Similarly, the decreased and neutral risk groups include at least one protective haplotype. Thus, similar groups in terms of haplotype makeup and risk do not significantly differ in gut bacterial composition. Additionally, the $R^2$ values of these comparisons show a gradient trend in the amount of variation explained by risk groups at opposite ends of the risk spectrum. For example, the greatest amount of variation is explained between the high and decreased risk groups ($R^2 = 0.00923$), followed by the high and neutral risk groups ($R^2 = 0.00824$), then the high and increased risk groups ($R^2 = 0.00805$). Because HLA risk explains little of the variation in the model, this suggests that the effect HLA has on the gut microbiome is modest and likely specific toward certain taxa. Also, comparisons between the highest risk subjects consistently explained the most amount of variation. This suggests that the risk-associated haplotypes together are more strongly driving compositional changes within the gut, whereas those without risk haplotypes or with at least one protective haplotype tend to be more similar in bacterial gut composition.

The PERMANOVA test is dependent upon distances of dissimilarity between samples. Previous work has shown that the choice of distance metric can have a drastic effect on the test result[20]. This is because different metrics test different hypotheses. Here the binomial distance measure was used as an improved alternative to the Bray–Curtis index. Other commonly used metrics applied to 16S rRNA amplicon data include Bray–Curtis and Jaccard. Bray–Curtis is considered quantitative, because it is mainly sensitive to highly abundant species, whereas

Jaccard is qualitative and considers the overlap of community members regardless of their relative abundances. The binomial distance metric is also quantitative but, unlike Bray–Curtis, includes joint absences, allowing pairs of samples missing the same ASV to appear more similar[21]. Differences in gut communities between risk groups were not significant using the Bray–Curtis distance (PERMANOVA: $R^2 = 0.00757$; F Model = 1.0445; *p*-value = 0.43), but significant differences were observed with the Jaccard distance (PERMANOVA: $R^2 = 0.00893$, F Model = 1.2257, *p*-value = 0.007). This suggests that the abundance of the bacterial ASVs observed may not be significantly influenced by HLA genetics. However, the presence or absence of certain bacteria are likely influenced by HLA as differences in community member overlap varies significantly by genetic risk.

An assumption of the PERMANOVA test is that all groups compared have similar variance (homogeneity of variance assumption), particularly in cases of uneven sampling between groups[22]. Beta dispersion was calculated based on genetic risk category adjusted for sampling bias, and an ANOVA was applied to test whether the average distance to the median variance was significantly different between groups. The result of the ANOVA showed that average distance to the median variance between groups was not significantly different (F value = 0.3928; *p*-value = 0.7583). Also, a Tukey HSD test showed that pair-wise tests of the variance assumption are not significant. Therefore, significance testing through PERMANOVA is not expected to be influenced by uneven dispersion among groups.

**Amplicon sequence variants associate with HLA genetic risk.** Because there are significant differences in microbiome composition between genetic risk groups, bacteria associated with

**Table 1 Risk groups that are significantly different by PERMANOVA**

|  | Df | SumsOfSqs | MeanSqs | F Model | $R^2$ | P value | P adjusted | Significance |
|---|---|---|---|---|---|---|---|---|
| *Binomial* |  |  |  |  |  |  |  |  |
| Risk_group | 3 | 20755 | 6918.4 | 1.3088 | 0.0097 | 0.01 | – | * |
| High vs. Decreased | – | – | – | 1.5698 | 0.00923 | 0.005 | 0.017 | * |
| High vs. Increased | – | – | – | 1.0236 | 0.00805 | 0.383 | 0.402 |  |
| High vs. Neutral | – | – | – | 1.354 | 0.00824 | 0.035 | 0.042 | * |
| Increased vs. Decreased | – | – | – | 1.3248 | 0.00554 | 0.035 | 0.038 | * |
| Increased vs. Neutral | – | – | – | 1.5178 | 0.00649 | 0.004 | 0.011 | * |
| Decreased vs. Neutral | – | – | – | 1.0601 | 0.00385 | 0.304 | 0.311 |  |
| *Bray–Curtis* |  |  |  |  |  |  |  |  |
| Risk_group | 3 | 1.169 | 0.38973 | 1.0445 | 0.00757 | 0.43 | – |  |
| High vs. Decreased | – | – | – | 1.1425 | 0.00657 | 0.204 | 0.225 |  |
| High vs. Increased | – | – | – | 0.6467 | 0.00499 | 0.964 | 0.973 |  |
| High vs. Neutral | – | – | – | 0.9827 | 0.00583 | 0.46 | 0.467 |  |
| Increased vs. Decreased | – | – | – | 1.3912 | 0.00569 | 0.066 | 0.077 |  |
| Increased vs. Neutral | – | – | – | 0.9212 | 0.00385 | 0.56 | 0.587 |  |
| Decreased vs. Neutral | – | – | – | 1.0117 | 0.00358 | 0.398 | 0.423 |  |
| *Jaccard (binary)* |  |  |  |  |  |  |  |  |
| Risk_group | 3 | 1.372 | 0.45718 | 1.2257 | 0.00893 | 0.007 | – | * |
| High vs. Decreased | – | – | – | 1.4114 | 0.00816 | 0.003 | 0.011 | * |
| High vs. Increased | – | – | – | 1.0449 | 0.00806 | 0.271 | 0.286 |  |
| High vs. Neutral | – | – | – | 1.3679 | 0.00816 | 0.012 | 0.016 | * |
| Increased vs. Decreased | – | – | – | 1.2179 | 0.00501 | 0.041 | 0.047 | * |
| Increased vs. Neutral | – | – | – | 1.3952 | 0.00586 | 0.002 | 0.008 | * |
| Decreased vs. Neutral | – | – | – | 0.9719 | 0.00346 | 0.551 | 0.555 |  |

PERMANOVA results after testing for significant differences in inter-subject distances using three metrics: Binomial, Bray–Curtis and Jaccard ($n = 403$ individual stool samples). Breastfeeding status at the time of sample collection was corrected for in the PERMANOVA model design. SumsOfSqs and MeanSqs refer to Sums of Squares and Mean Squares, respectively

genetic risk across the entire dataset were identified. Using Linear discriminant analysis Effect Size (LEfSe) after grouping ASVs at the genus level, significant associations between high risk subjects and members of the Saccharimonadaceae family were identified (Fig. 2a, b). Additionally, the family Erysipelotrichaceae was associated with subjects at increased genetic risk (Fig. 2a, c). Interestingly, two genera of the family Peptostreptococcaceae, specifically *Intestinibacter* and *Romboutsia*, were associated with neutral risk HLA (Fig. 2a, d). Differential feature plots show that the taxa described above have higher average relative abundance in their associated risk group than any other group (Fig. 2b–d). Overall, taxa found to be associated with a particular risk group through LEfSe had higher average relative abundance in their associated risk group. Because subjects at neutral risk often have either DR3 or DR4 plus at least one protective haplotype, the protective haplotypes may be important for this association, though no associations with the decreased risk group were observed in the entire dataset level with LEfSe.

DESeq2 was used to find ASVs associated with genetic risk through pair-wise comparisons (Table 2). Again, *Intestinibacter* and *Romboutsia* were found to be associated with neutral risk compared with the high-risk group. Comparing the high (DR3/DR4 only) vs. decreased risk groups revealed that *Intestinibacter* and *Romboutsia* were also associated with decreased risk. This suggests that DR3 and DR4 together have a greater impact on the association of these ASVs than either haplotype alone (as in the increased risk group), and that the presence of protective haplotypes may also be important for their presence and/or abundance. Also, when comparing the increased vs. neutral groups, *Bifidobacterium* was associated with neutral risk with a relatively high base mean compared to other associated ASVs found by DESeq2, suggesting this ASV was more abundant across all samples compared to other identified significant ASVs. Even though *Escherichia/Shigella* members were shared within both the neutral and increased risk groups, they represent unique ASVs and therefore, have the potential to be unique strains. *Klebisella*

and *Veillonella* ASVs were consistently associated with high risk (DR3/4) (Table 2).

DESeq2 was also able to identify bacterial ASVs associated with particular HLA genotypes or haplotypes (e.g., heterozygous DR3/DR4, or genotypes positive for DR3 or DR4 without protective haplotypes). *Intestinibacter* and *Romboutsia* were associated with the DR3 positive increased risk genotype over the DR3/DR4 group. However, neither ASV was associated when comparing DR4 positive with DR3 positive increased risk genotypes, suggesting that the high-risk heterozygous genotype may be exerting a selective pressure against these bacteria. Interestingly, when comparing genotypes where DR3 or DR4 positive haplotype is associated with a protective haplotype, these two ASVs are associated with the absence of DR3 and at the same time, are also not associated with DR4. Because different haplotype combinations will lead to varying degrees of antigen-binding affinity, associations between HLA genotype and bacteria within the gut are likely to be genotype-specific. This is evident in that either having or lacking a protective haplotype in combination with DR3 associates with a different group of gut flora.

**Prevalence analysis reveals trends by limiting noise.** Amplicon 16S data is sparse and therefore populated with numerous low or zero counts for ASVs that are rarely seen. These rare taxa could be considered noise, as they may not be relevant to the biological question because they appear in very few subjects. To limit this noise, we filtered these data using a method that considers ASV prevalence. ASVs were filtered at 5% increments for the entire dataset (Table 3). A prevalence cutoff of 45% was chosen to assess those ASVs which were present in nearly half of all individuals analyzed. Higher prevalence cutoffs, up to 75%, could be obtained for the full dataset at the cost of additional ASVs. This filtering resulted in clear separation by PCoA between genetic risk groups that could not be observed from the raw dataset, where more distinct clusters form between ASVs prevalent in 45% of each risk

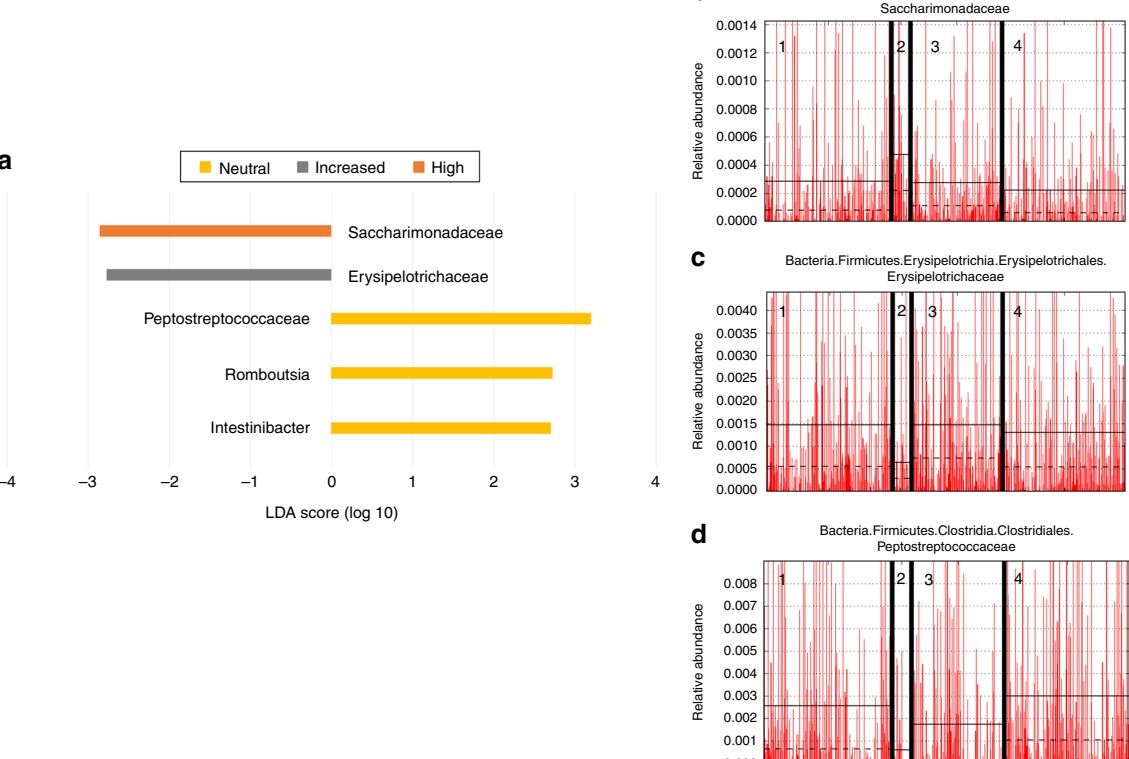

**Fig. 2** Specific taxa are associated with genetic risk. **a** LEfSe biomarker discovery results for the entire dataset ($n = 403$ individual stool samples). Significant taxa are displayed according to their associated risk group and their differential $\log_{10}$ LDA score. **b**–**d** Differential features identified by LEfSe have higher average relative abundance with their associated risk group. **b** Saccharimonadaceae, **c** Erysipelotrichaceae, **d** Peptostreptococceae, (1) Decreased risk, (2) High risk, (3) Increased risk, (4) Neutral risk. Breastfeeding status at the time of sample collection was corrected for in the LEfSe model design. Source data are provided in the source data file

group (Fig. 3), regardless of whether those samples from participants still breastfeeding at the time of sample collection were included (Fig. 3a) or not (Fig. 3b). In other words, those ASVs that are present more frequently (in at least 45% of subjects in this dataset) make up distinct patterns of composition depending on HLA-driven genetic risk for T1D.

Considering only those ASVs that are present in at least 45% of subjects in each risk group, 40 unique ASVs remain from the total 4450 ASVs in the raw dataset. Of the 40 total ASVs, 14 are prevalent among all risk groups, yet the other 16 are shared among 2 or 3 groups, while others still are unique to only one group (Fig. 4). Though, at the genus level, some taxa are shared, these organisms are represented by unique sequences and should therefore be considered unique organisms. In support of the differential abundance results, ASVs that are members of *Intestinibacter* and *Romboutsia* are most prevalent in the neutral and decreased HLA-risk groups. This provides further evidence that protective HLA haplotypes may be important for shaping the composition of bacteria in the human gut. Interestingly, an ASV belonging to the *Bifidobacterium* genus was prevalent among all risk groups. However, another *Bifidobacterium* ASV was prevalent just among high and increased risk subjects, while yet another was unique to just those at neutral risk. This highlights the importance of pairing ASV-level sequence resolution with prevalence, since the distinction among these unique sequences would be overshadowed by rare sequences and overlooked at the taxonomic rather than the single nucleotide variant level.

**Geographical clustering identifies bacterial hotspots**. Geography explains a significant amount of variation between

subjects' gut microbiomes in ABIS (PERMANOVA: $R^2 = 0.03651$; F Model $= 1.3503$; $p$-value $= 0.001$). Differences in geography can include underlying covariates such as diet and other lifestyle factors, and the effects of these factors can be exacerbated with increasing geographical distance[23]. Although the ABIS cohort benefits from geography confined to southeast Sweden, the significant impact that geographical factors have on subjects' gut microbiomes is still evident through PERMANOVA. To limit this impact, towns were divided into three distinct clusters: the northeastern region (Linköping and Norrköping), the central region (Jönköping, Nässjö, Gislaved and Värnamo), and the southern region (Kalmar and Karlskrona). The towns within these regions were chosen for clustering based on their geographical proximity and because they provide the largest number of samples in the dataset, including many of those from high-risk subjects.

This geographical clustering, paired with differential abundance and prevalence analysis, allowed us to determine at which sites associations within the entire dataset were likely to have originated. LEfSe was again able to identify taxa associated with genetic risk at each cluster and overall among the clusters (Fig. 5). The LEfSe results show that among all clusters, *Intestinibacter* remains associated with neutral risk. Furthermore, this association was identified in the northeastern region while *Romboutsia* was associated with decreased risk in this region. In the southern region, *Romboutsia* was associated with neutral risk while *Intestinibacter* was associated with decreased risk. DESeq2 results also confirm that ASVs belonging to *Intestinibacter* and *Romboutsia* were associated with neutral and decreased HLA groups in the northeastern and southern regions.

**Table 2 DESeq2 supports LEfSe results at the ASV level**

| Pair-wise | base mean | Log2 fold change | lfcSE | stat | *p* value | *p* adjusted | Genus |
|---|---|---|---|---|---|---|---|
| High | 33.90 | 11.12 | 2.85 | 3.90 | 9.74E-05 | 0.0093 | *Klebsiella* |
| | 16.16 | 23.66 | 5.92 | 4.00 | 9.74E-05 | 0.0067 | *Veillonella** |
| | 2.59 | 23.04 | 5.92 | 3.89 | 9.93E-05 | 0.009345 | *Veillonella** |
| Decreased | 68.40 | −2.89 | 0.84 | −3.45 | 0.0005576 | 0.03653 | *Romboutsia* |
| | 40.17 | −4.74 | 1.24 | −3.83 | 0.000128 | 0.011178 | *Enterococcus* |
| | 36.80 | −3.37 | 0.75 | −4.49 | 7.13E-06 | 0.001006 | *Intestinibacter* |
| High | 91.95 | 19.79 | 5.84 | 3.39 | 0.0007063 | 0.023203 | *Veillonella** |
| | 33.90 | 9.69 | 2.87 | 3.37 | 0.0007523 | 0.024312 | *Klebsiella* |
| | 16.16 | 35.34 | 5.97 | 5.92 | 3.14E-09 | 5.95E-07 | *Veillonella** |
| Neutral | 68.40 | −2.88 | 0.84 | −3.40 | 0.0006627 | 0.021951 | *Romboutsia* |
| | 40.17 | -4.42 | 1.25 | −3.55 | 0.0003914 | 0.013917 | *Enteroccocus* |
| | 36.80 | −3.86 | 0.76 | −5.10 | 3.43E-07 | 4.40E-05 | *Intestinibacter* |
| High | 91.95 | 23.27 | 5.98 | 3.89 | 9.97E-05 | 0.003386 | *Veillonella* |
| | 33.90 | 11.17 | 2.94 | 3.79 | 0.0001484 | 0.00458 | *Klebsiella* |
| | 17.77 | 13.94 | 4.43 | 3.15 | 0.0016554 | 0.035626 | *Phascolarctobacterium* |
| Increased | 40.17 | −4.61 | 1.27 | −3.62 | 0.0002916 | 0.007681 | *Enterococcus* |
| | 20.27 | −2.43 | 0.64 | −3.81 | 0.0001388 | 0.004329 | *Lachnoclostridium* |
| | 17.55 | −2.33 | 0.63 | −3.72 | 0.0002009 | 0.005902 | *Erysipelatoclostridium* |
| Increased | 9.15 | 30.00 | 3.85 | 7.78 | 7.03E-15 | 3.26E-12 | *Escherichia/Shigella** |
| | 6.86 | 16.40 | 3.86 | 4.25 | 2.10E-05 | 0.000638 | *Escherichia/Shigella** |
| | 5.84 | 9.63 | 2.42 | 3.98 | 6.90E-05 | 0.002038 | *Bacteroides* |
| Decreased | 19.75 | −19.63 | 3.86 | −5.09 | 3.53E-07 | 2.00E-05 | *Veillonella* |
| | 13.36 | −29.79 | 3.86 | −7.72 | 1.12E-14 | 4.15E-12 | *Megasphaera* |
| | 6.97 | −6.55 | 1.70 | −3.85 | 0.0001173 | 0.003287 | *Klebsiella* |
| Increased | 14.13 | 7.69 | 1.71 | 4.49 | 6.96E-06 | 0.000299 | *Citrobacter* |
| | 5.84 | 10.89 | 2.44 | 4.47 | 8.00E-06 | 0.00034 | *Bacteroides* |
| | 3.06 | 27.04 | 3.89 | 6.95 | 3.61E-12 | 3.02E-10 | *Escherichia/Shigella* |
| Neutral | 267.32 | −2.30 | 0.68 | −3.39 | 0.0007071 | 0.026723 | *Bifidobacterium* |
| | 19.75 | −16.58 | 3.89 | −4.26 | 2.01E-05 | 0.000822 | *Veillonella* |
| | 6.97 | −7.24 | 1.71 | −4.23 | 2.38E-05 | 0.000964 | *Klebsiella* |
| Neutral | 9.15 | 18.37 | 3.56 | 5.16 | 2.48E-07 | 3.18E-05 | *Escherichia/Shigella** |
| | 6.86 | 15.09 | 3.56 | 4.24 | 2.24E-05 | 0.001983 | *Escherichia/Shigella** |
| | 5.81 | 14.73 | 3.56 | 4.14 | 3.50E-05 | 0.002961 | *Anaeroglobus* |
| Decreased | 13.36 | −28.66 | 3.56 | −8.05 | 8.41E-16 | 8.83E-13 | *Megasphaera* |
| | 6.20 | −12.57 | 3.56 | −3.53 | 0.0004112 | 0.02818 | *Sarcina* |
| | 2.59 | −27.94 | 3.56 | −7.84 | 4.33E-15 | 3.45E-12 | *Veillonella* |

The top three ASVs by base mean from each group in the pair-wise comparison are listed and described taxonomically at the genus level ($n = 403$ individual stool samples). Genera indicated with an asterisk (*) represent bacteria shared at the taxonomic level but are actually unique sequences. Breastfeeding status at the time of sample collection was corrected for in the DESeq2 model design. Source data are provided in the source data file

**Table 3 Prevalence filtering focuses analysis on most shared ASVs**

| Prevalence cutoff (%) | Out of bag error rate | ASV count | Number of sequences |
|---|---|---|---|
| 0 | 0.691 | 4450 | 7610000 |
| 5 | 0.45544554 | 532 | 3572138 |
| 10 | 0.18316832 | 262 | 3274985 |
| 15 | 0.19306931 | 183 | 3091195 |
| 20 | 0.13861386 | 138 | 2842684 |
| 25 | 0.10148515 | 107 | 2588149 |
| 30 | 0.10148515 | 85 | 2351261 |
| 35 | 0.09158416 | 70 | 2110874 |
| 40 | 0.07920792 | 49 | 1808389 |
| 45 | 0.0470297 | 40 | 1499585 |
| 50 | 0.13366337 | 28 | 1242599 |
| 55 | 0.25742574 | 23 | 1196297 |
| 60 | 0.16584158 | 16 | 1087380 |
| 65 | 0.13613861 | 13 | 1010804 |
| 70 | 0.34405941 | 8 | 930300 |
| 75 | 0.14108911 | 7 | 860687 |

Results of calculating prevalence levels for the entire dataset ($n = 403$ individual stool samples) in 5% increments. A prevalence cutoff of 45% was chosen to focus on the most commonly shared set of 40 ASV sequences

Prevalence filtering was also applied to each geographical region separately by the same prevalence level used for the entire dataset (45%). The prevalence filtered regions show distinct patterns of separation by HLA risk group through PCoA (Fig. 6). Variation among the most prevalent ASVs by risk group is most considerable in the Southern region, though overlap with these ASVs can be seen between the high/increased and decreased/ neutral risk groups, respectively. In the Northeastern region, overlap between gut bacterial communities is more consistent across all risk groups. Meanwhile, in the Central region, a high degree of overlap exists between all risk groups aside from the highest risk, which are substantially separated, indicating that the variation in community overlap among the most prevalent bacteria in this risk group is strong, regardless of breastfeeding status (Fig. 6d). In other words, each genetic risk group comprises a distinct combination of ASVs that are present in most of the participants at each risk level. This would suggest that the ASVs forming these highly prevalent clusters are those most influenced by host HLA genotype, posing a potential mechanism of selection toward those bacteria in the gut.

**Bifidobacteria viable after storage at −80 °C for 19 years.** Sample integrity is an important consideration when working with samples that have been stored for decades. Here, we assessed

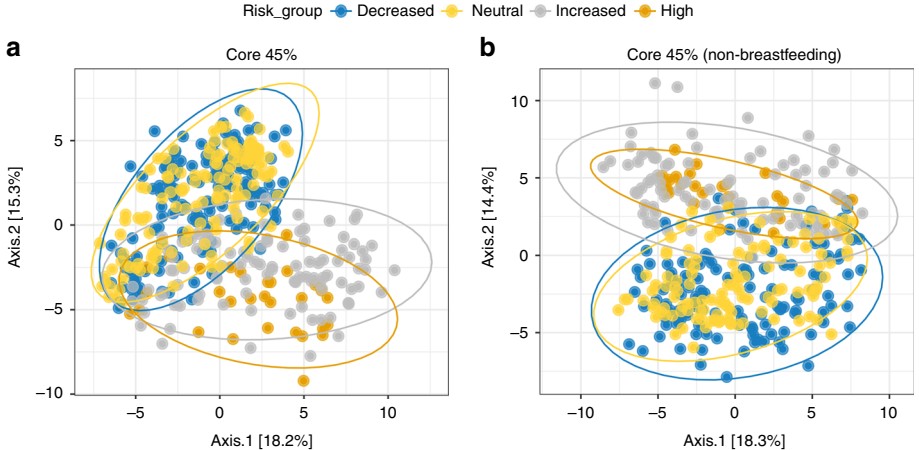

**Fig. 3** Prevalence uncovers unique clusters among the most shared ASVs by risk. **a, b** PCoA depicting inter-subject distances based on the binomial metric at 45% prevalence, both including participants breastfed at sample collection (**a**) and without participants breastfed at sample collection (**b**). Ellipses for each risk group are calculated based on a 95% confidence for each group

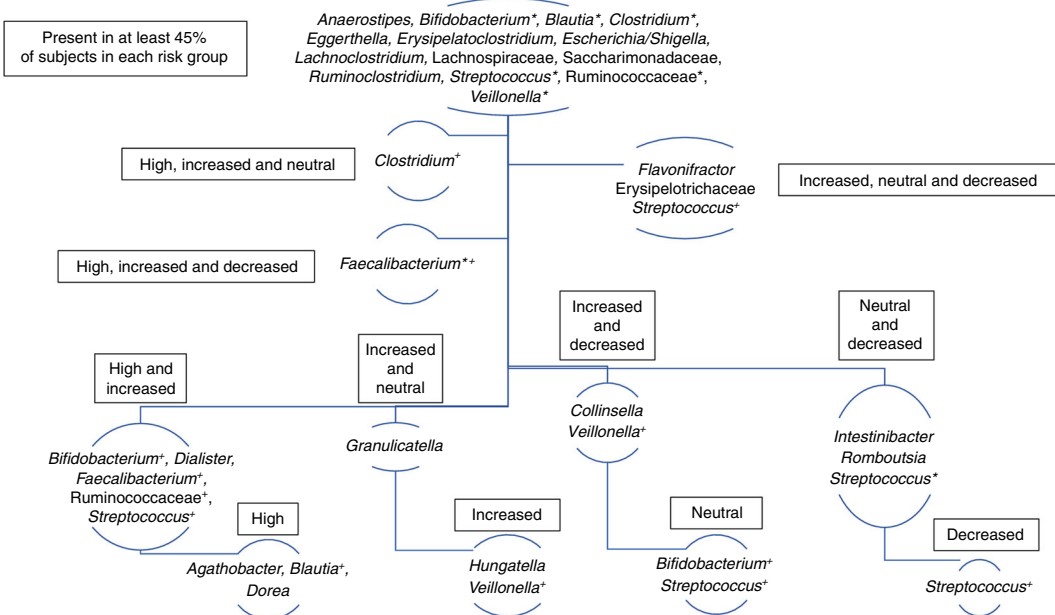

**Fig. 4** Highly prevalent ASVs are either shared or unique to HLA groups. Hierarchy organization map displaying those taxa that are present in at least 45% of subjects in one or more risk groups. Taxa indicated with an asterisk (*) are represented by more than one ASV which are considered unique. Furthermore, those taxa indicated with a cross (+) represent bacteria shared at the taxonomic level but are actually unique to one or a combination of groups at the ASV level

the viability of −80 °C frozen stool samples through isolation of non-spore forming, facultative anaerobic strains of *Bifidobacterium*. Members of the *Bifidobacterium* genus are considered relatively sensitive to long-term storage conditions, since they are non-spore forming and are sensitive to oxygen exposure. Nevertheless, strains of two *Bifidobacterium* species, *B. breve* and *B. longum*, were isolated from a stool sample stored at −80 °C for 19 years using media selective for the genus.

## Discussion

HLA gene alleles appear to have a significant effect on the bacterial composition of the late infant gut based on our findings from children in the ABIS cohort. This effect is important because of the many implications with how genetic pre-disposition to

autoimmune disorders might determine environmental factors, such as the microbiome, that ultimately may lead to disease progression. In the case of T1D, this is the first time to our knowledge that HLA genetic risk for developing autoimmunity has been associated with distinct changes in the gut microbiome in a general population human cohort. Beyond just genetic risk for T1D, our results implicate specific HLA genotypes with distinct changes in members of the gut community, which highlights an important understanding of the interaction and influence of host genetics on the human microbiome. How HLA genetics might lead to environmental triggers for autoimmune disease originating in the gut deserves further investigation on many fronts.

The greatest advantage to the ABIS cohort is its general population design. Because the general population is sampled and

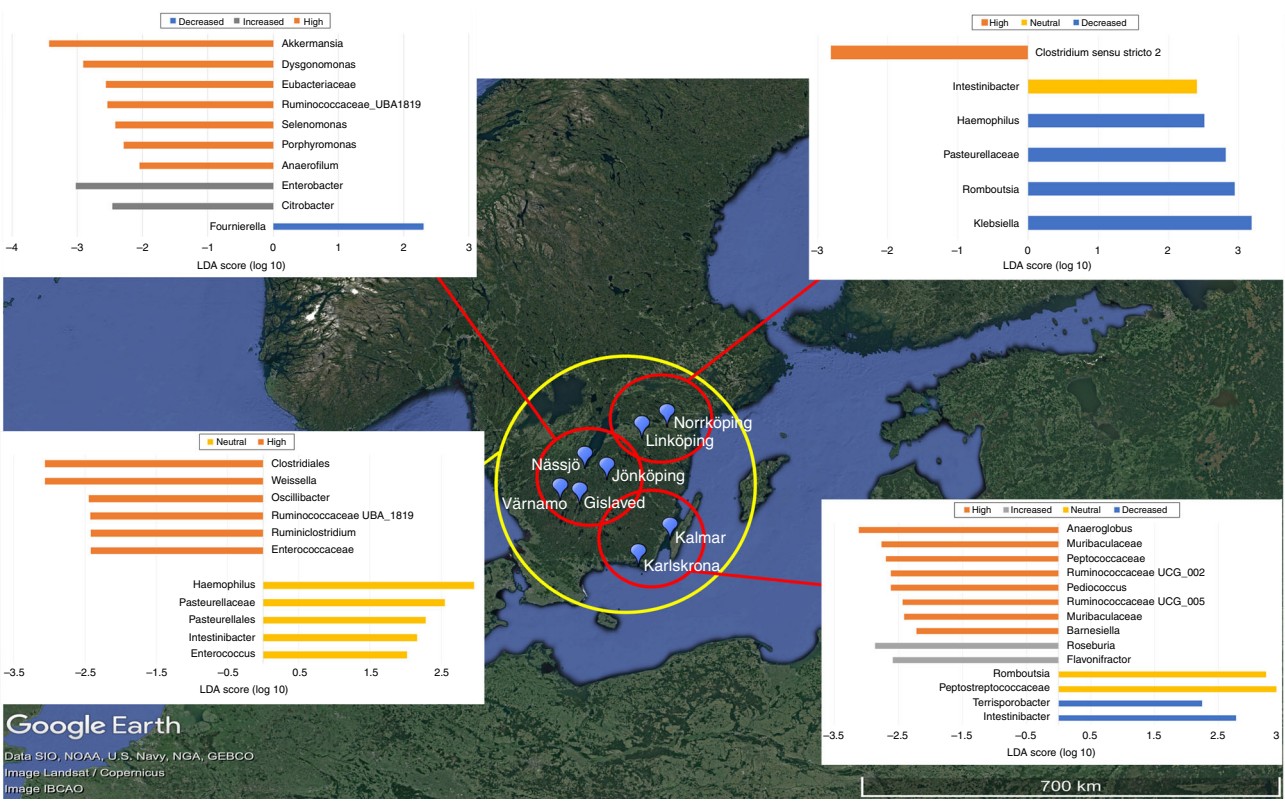

**Fig. 5** LEfSe identifies sources of ASVs by geographical region. Linear discriminant analysis Effect Size (LEfSe) is able to identify taxa as biomarkers associated with genetic risk groups in the three geographical regions and among all three regions. Northeastern region $n = 140$ individual stool samples, Southern region $n = 65$ individual stool samples, Central region $n = 160$ individual stool samples. Map data © 2019 Google, SIO, NOAA, U.S. Navy, NGA, GEBCO. Image © 2019 Landsat/Copernicus. Source data are provided in the source data file

not just those at high genetic risk for T1D, HLA associations with the gut microbiome can be tested across a spectrum of genotypes. High genetic risk for autoimmunity likely obscures potential effects from the environment that can be identified by inclusion of all genetic types as we see with ABIS. Additionally, geography is known to be a potent source of variation between subjects[13]. Like the DIPP cohort in Finland, ABIS benefits from a relatively confined geographic sampling design restricted to southeast Sweden[12]. Unlike other studies which have historically relied on operational taxonomic unit (otu) clustering analysis, this work has benefited from higher sequence resolution through the use of ASVs[24]. At the ASV level, sequences may share the same taxonomy at the genus or species levels while being distinct at the sequence level. This implies that bacteria of the same name may actually be functionally distinct within or between subject groups and should be treated as such. In addition, taking into account the prevalence of each ASV within risk groups has allowed us to identify potentially important bacteria in the gut that would have been overlooked by examining an entire dataset with all its noise.

Recent findings from the TEDDY study show that development of insulin autoantibodies (IAA)-only is associated with DR3/4 and DR4/4 HLA genotypes around 1 year of age, while development of glutamic acid decarboxylase autoantibodies (GADA)-only is associated with DR3/3[25]. In addition, previous work in Finland has shown that insulinoma associated antigen-2 autoantibodies (IA-2A)-only is associated with (DR4)-DQA1*03-DQB1*0302[26]. This would suggest that the development of autoimmunity is directly impacted by host HLA genetics. However, the results presented here also implicate HLA with specific changes in gut bacterial composition relative to DR3/4 heterozygosity and genotypes positive for either DR3 or DR4. Therefore,

the question persists whether the association between HLA and the route of autoimmunity is primarily genetic or is caused by secondary changes in the bacterial gut environment that are mediated by genetic risk. Nevertheless, these findings contribute to our understanding of the interaction between environmental exposures in the gut and host genetics in disease development.

Both differential abundance and prevalence analysis indicate that two members of the family Peptostreptococcaceae, *Intestinibacter* and *Romboutsia*, are consistently associated with lower genetic risk HLA genotypes compared to DR3/DR4. Not only does this hint at a potential for conserved functions, which may be beneficial in preventing T1D, but also the possibility of a conserved bacterial antigen recognized by higher risk genotypes that leads to the depletion of these taxa in the gut. This selection would be afforded by polymorphisms in the antigen-binding groove of the class II MHC molecule that will ultimately alter the binding affinity of the MHC molecule toward a subset of bacterial antigens. Effects of MHC polymorphism on the binding of bacterial superantigens or toxins, non-processed molecules binding to non-conventional binding sites of MHC and TCR molecules, has also been observed[5]. Many studies have implicated class II MHC genetics in the shaping of the gut microbiome[6–8,27]. However, this work was done using animal models including mainly transgenic mice. Thus, the results presented here show that HLA genotype is linked to unique changes in the human gut microbiome in a general population cohort.

Members of the order Clostridiales have recently undergone reclassification in the light of phenotypic and phylogenetic data[28]. *Intestinibacter bartlettii*, formerly *Clostridium bartlettii*, was reclassified at the same time the genus *Romboutsia* was described (with the isolation of *Romboutsia ilealis*)[29,30]. *I. bartlettii* is the

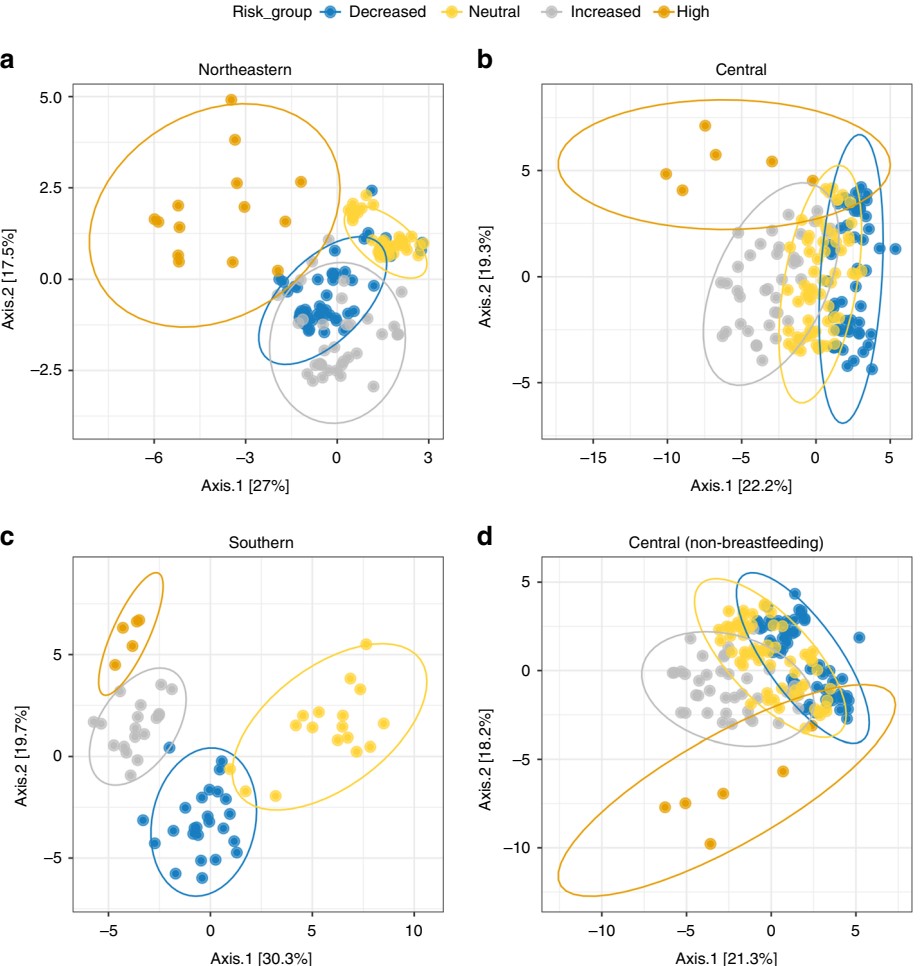

**Fig. 6** Prevalent ASV clusters vary by HLA genotype across geography. **a–d** PCoA of inter-subject binomial distances for each of the three geographical clusters after applying 45% prevalence filter: **a** the Northeastern, **b** the Central, and **c** the Southern region. **d** Central region 45% prevalence after removal of samples from participants still breastfed at the time of sample collection. Ellipses are calculated based on a 95% confidence for each group

only species currently in the *Intestinibacter* genus after its reclassification from *Clostridium* and is yet to be represented by a single complete genome sequence in a public database. Little has been described about *Intestinibacter* spp. other than a decreased abundance associated with administration of metformin (a type 2 diabetes (T2D) drug) and a negative correlation with insulin resistance in T2D[31,32]. The association of *Intestinibacter* with lower genetic risk for autoimmunity requires further work to understand what role these organisms play in the gut and how they may be selected against by high-risk HLA.

Probiotics have great potential as a means to modulate changes in the host microbiome to prevent or ameliorate a number of different diseases including T1D autoimmunity. Both mouse and rat models have implicated lactic acid bacteria (e.g. *Lactobacillus* spp.) as potential probiotics for the prevention of T1D[33,34]. Administration of a multi-species probiotic (including Lactoba-cilli) has been shown to be beneficial in treating acute pancreatitis in a rat model by reducing both bacterial translocation to the pancreas and bacterial overgrowth of potential pathogens in the duodenum[35,36]. Probiotic administration was associated with an increase in abundance of a new bacterial phylotype referred to as CRIB (commensal rat ileum bacterium) which was later described as *Romboutsia ilealis*[30,37]. The increased abundance of *R. ilealis* was also associated with lower plasma levels of pro-inflammatory cytokines[37]. As the authors speculate, it is possible that the administration of the probiotic mixture worked indirectly by

stimulating the growth of *R. ilealis* in the ileum. A study conducted in mice demonstrated that colonization with certain Clostridia led to decreased mucosal erosion and increased IL-10 producing regulatory T cell accumulation in the colon. These functions within the host could be important in preventing the development of autoimmunity. The role *Romboutsia* plays in mediating these effects is unknown.

The effects HLA genetics have on shaping the host microbiome could be important for other autoimmune diseases besides T1D where HLA is the main genetic component. Like T1D, much of the genetic risk for developing celiac disease autoimmunity (CDA) is conferred by the class II HLA region[38]. Similarly, environmental influences are thought to be important as not all those who are genetically susceptible develop the disease upon exposure to dietary gluten, the main trigger of CDA. Unlike T1D, the greatest genetic risk for developing CDA is attributable to the (DR3)-DQA1*05-DQB1*02 haplotype[39]. Because the level of genetic risk for CDA and T1D differ by HLA genotype, in addition to differences in the localization of the pathology (small intestine in CDA vs. pancreas in T1D), the etiology of the two diseases likely vary. Still, shaping of the host intestinal micro-biome through an HLA mechanism could play an important role in not just T1D but in CDA and other autoimmune disorders.

The association between HLA risk alleles for T1D and the gut microbiome is shown here using a general population cohort from southeastern Sweden. Three methods were used to

consistently identify specific bacterial taxa and 16S rRNA sequences (ASVs) associated with genetic risk. These results have implications that go far beyond the gut microbiome. They suggest that HLA alleles can shape the subject's ability to interact with the environment. Thus, HLA risk alleles can mask potentially co-occurring environmental influences so strongly that environmental effects can be difficult to observe in the those T1D cohorts designed to include only subjects at high genetic risk for T1D. Indeed, previous studies exploring the effects of other environmental factors such as vitamin D, breast feeding duration, probiotic use, gluten introduction and virus infection may also be masked by increased genetic risk. Only a large, general population study may be able to evaluate the full impact of these environmental influences and their interactions with genetic risk for T1D.

The All Babies in Southeast Sweden (ABIS) cohort is ideal for investigating the influence of HLA genetic risk on the microbiome since it was designed to survey the general population, making up a diverse range of HLA genotypes. The information gained from this type of cohort furthers our understanding of how HLA genetic risk drives changes in the gut microbiome and how genetics may be "setting the stage" for environmental triggers that ultimately lead to T1D autoimmunity. Furthermore, the results obtained in this study can likely be extended to investigate other autoimmune diseases where HLA serves as the primary genetic risk factor.

## Methods

**Sample collection**. This study is based on the ABIS cohort (All Babies in Southeast Sweden), a prospective population-based cohort study including all children born in southeast Sweden during the period 1 October 1997–1 October 1999. The parents gave their informed consent after oral and written information and possibility of obtaining information via video film. Participating mothers completed questionnaires both at birth and at 1 year of age for the infant and diaries were kept during the first year of life[40]. Information collected in the questionnaires and diaries include, but are not limited to, antibiotic use, duration of breast feeding, and more.

Stool samples were collected from the diaper of the infant by use of a sterile spatula and tube provided by the WellBaby Clinic. The sample tubes were labeled with a unique subject identifier and frozen immediately after collection, either at the infant's home or at the clinic. Samples collected at home were transported frozen, using freeze clamps, to the WellBaby Clinic. After arrival at the clinic, the samples were stored dry at −80 °C. All 403 stool samples used in this analysis were collected at 1 year of age, and each subject is represented by a single one-year sample.

**Institutional Review Board approvals**. The ABIS-study has ethical approvals from the Research Ethics Committees of the Faculty of Health Science at Linköping University, Sweden, Ref. 1997/96287 and 2003/03-092 and the Medical Faculty of Lund University, Sweden (Dnr 99227, Dnr 99321). All parents of the children in the ABIS-study gave their informed consent after careful oral and written information in addition to video film presentation. The microbiome analysis performed at the University of Florida was approved by the University of Florida's Institutional Review Board as an exempt study assigned as IRB201800903.

**HLA genotyping**. HLA-DR/DQ genotypes associated with risk and protection were defined using typing for HLA-DQB1 and informative -DQA1 and DRB1 alleles for deducing presence of common European HLA-DR-DQ haplotypes variously associated with disease risk using sequence specific hybridization with lanthanide labelled oligonucleotide probes[41,42]. To enable the comparison of different levels of HLA risk for developing T1D autoimmunity, subjects with available HLA genotype data were placed into one of four categories of risk based on their HLA genotype. Subjects at the highest genetic risk for developing autoimmunity are represented by a single HLA genotype in this dataset, consisting of both increased risk-associated haplotypes: (DR3)-DQA1*05-DQB1*02 and (DR4)-DQA1*03-DQB1*0302 (DR3/4). In contrast, those at lowest risk are denoted by the absence of these haplotypes and the presence of one or two protective haplotypes (DR15)-DQB1*0602, (DR13)-DQB1*0603, (DR5)-DQA1*05-DQB1*0301 and (DR7)-DQA1*0201-DQB1*0303. Subjects having either (DR3)-DQA1*05-DQB1*02 or (DR4)-DQA1*03-DQB1*0302 without presence of protective haplotypes were defined to be at increased risk. Those with (DR3)-DQA1*05-DQB1*02 or (DR4)-DQB1*0302 and one of the protective haplotypes and those without any risk or protective haplotypes are at neutral genetic risk. A subset of subject characteristics for the samples used in this work is described in Supplementary Table 1.

HLA genotype information for the subjects included in this study is provided in the source data file.

**DNA Extraction and 16S rRNA barcoded PCR**. DNA from each sample was extracted from ~200 mg of stool[12], using the E.Z.N.A Stool Extraction Kit following the manufacturer's protocol (Omega Bio-tek, Doraville, CA). Samples were randomized to prevent the introduction of bias during extraction and blank negative controls were introduced alongside the samples to verify the absence of contamination in the extraction kit components. DNA used for subsequent PCR was quantified and assessed for purity using a Nanodrop spectrophotometer (Thermo Scientific, Wilmington, DE). Previously used custom barcoded primers 341F and 806R, targeting the V3-V4 variable regions of the 16S rRNA gene, were employed and PCR amplification was carried out with the following modifications. Briefly, 50 nanograms of extracted DNA was used as template in a final reaction volume of 50 μl containing 25 μl of 2X GoTaq Colorless Master Mix (Promega, Madison, WI), 10 μM of each primer, 0.1 μg/μl BSA, brought to final volume with nuclease free water. The following cycling conditions were used for amplification: initial denaturation at 95 °C for 2 min, followed by 30 cycles of 95 °C for 20 s, 61 °C for 30 s, 72 °C for 30 s, and a final elongation step of 72 °C for 5 min. All PCR reactions were run on 1% agarose gels to verify correct amplification and purified using the E.Z.N. A Cycle Pure kit per the manufacturer's instructions (Omega Bio-tek, Doraville, CA). The purified PCR products were then quantified using the 1X dsDNA High Sensitivity kit with a Qubit 2.0 fluorometer (Invitrogen, Life Technologies Inc., Carlsbad, CA). An equal mass of amplicons from each sample were pooled and sequenced on the Illumina MiSeq platform (ICBR, Gainesville, FL).

**V3-V4 16S sequencing using Illumina Miseq 2 × 300bp**. Five Illumina MiSeq flow cells were used to generate 67 gigabases of nucleotide sequencing data for 965 samples. Sequencing read pre-processing, including merging and demultiplexing, was done using scripts available through Qiime v1.9.1[43]. Forward and reverse sequencing reads were merged based on overlap to generate single reads using fastq-join https://github.com/ExpressionAnalysis/ea-utils. The joined reads were analyzed for quality using FastQC https://github.com/s-andrews/FastQC. The forward primer sequence used during PCR was trimmed using the fastx_trimmer tool available in the FASTX-Toolkit (http://hannonlab.cshl.edu/fastx_toolkit). The reads were then labelled according to 11-base barcodes located at the 3′ end of the read. Finally, the labelled reads were demultiplexed into separate FASTQ files by sample ID.

**Sequencing read processing into amplicon sequencing variants**. Demultiplexed sequences were further processed into amplicon sequencing variants (ASVs) using the DADA2 software package available in R[44] (https://www.R-project.org). Briefly, the demultiplexed FASTQ files were visually assessed for quality via the "plotQualityProfile" function. Read quality dropped sharply beyond 400 nucleotides and the reads were truncated to this length. The truncated reads were also filtered to allow for no ambiguous nucleotides (N), a maximum expected error rate of 2, further truncation after encountering a base with Q-score 2 and the removal of PhiX reads. Chimeric sequences were filtered using the consensus method and taxonomy was assigned using the SILVA_v132 training set[45–48]. The resulting ASV table was used to analyze the composition of the stool microbiome using the phyloseq package[49].

Further filtering removed samples with fewer than 1000 reads and samples with no subject genotype information. The remaining samples were rarefied to an even sequencing depth of 10,000 reads per sample, a sufficient sequencing depth verified through rarefaction curves. After filtering, 603 of the total 965 samples with HLA genotype data remained for analysis. Because not all genetic risk groups are represented in every town sampled, only those towns with at least one subject from each risk group were retained, resulting in 403 samples for analysis. The number of subjects in each risk group category are described in Supplementary Table 1.

**Statistical analysis of stool microbiome**. Differentially abundant ASVs between the genetic risk groups were determined using DESeq2 and LEfSe[50,51]. Alpha diversity (including plots) were calculated using the Microbiome R package (http://microbiome.github.com/microbiome). The Microbiome package was also used to calculate and filter by ASV prevalence. Differences in microbiome composition by genetic risk and other covariates were tested for using the permutational multivariate analysis of variance (PERMANOVA) test through the "adonis" function in the vegan R package (https://github.com/vegandevs/vegan). All PERMANOVA tests were performed with the default 999 permutations. PERMANOVA was used to test for significant differences in distance dissimilarity on a number of covariates including duration of breastfeeding, breastfeeding status at sample collection, antibiotic usage in the first year and within one month of sample collection (yes/no), mode of delivery and gender. The vegan package was also used for assessing multivariate homogeneity of variance (beta dispersion) between sampling groups using PERMDISP2. For calculating presence/absence metrics such as Jaccard, the "binary = TRUE" option was set. P-value results from statistical testing were corrected for false discovery rate (FDR) using the Benjamin–Hochberg (B–H) method[52]. Principal coordinate analysis (PCoA) plots, including the calculated 95% confidence ellipses, were generated using the ggplot2 R package[53].

**Culturing and identification of ASV isolates**. Aliquots of stool were stored at −80 °C prior to DNA extraction to culture bacteria of interest, such as those associated with health and low genetic risk. Samples showing the highest relative abundance of significant *Bifidobacterium* ASVs were serially diluted in 1X phosphate buffered saline and plated on pre-reduced Modified *Bifidobacterium* Agar (Becton, Dickinson and Company) without the addition of lactulose. Individual colonies were isolated from the two lowest dilution plates and culture stocks were stored in a 15% glycerol solution at −80 °C. DNA from each isolate was extracted using the E.Z.N.A Bacterial DNA kit (Omega Bio-tek, Doraville, CA) and the full length 16 S rRNA gene sequence was amplified by PCR using universal primers 27F and 1492R[54]. The resulting PCR products were Sanger sequenced to determine the identity of the isolates (Eton Bioscience, Inc., San Diego, CA). Resulting sequences were compared to the GenBank nucleotide database for identification using BLAST[55].

## Data availability

The paired-end 16S raw sequencing data generated in this study is available through the NCBI Sequence Read Archive under BioProject PRJNA510423. The source data underlying Figs. 1c, 2, 5 and Table 2 are provided as a source data file. Also, the HLA genotypes and associated sample metadata used for statistical comparison are available in the source data file.

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

## Acknowledgements

We thank all children and their parents in the ABIS study, and the staff working with collection of questionnaires and biological samples in ABIS. ABIS was supported by Barndiabetesfonden (Swedish Child Diabetes Foundation); Swedish Council for Working Life and Social Research, Grant/Award Numbers: FAS2004-1775, FAS2004–1775; Swedish Research Council, Grant/Award Numbers: K2005-72 × -11242-11A and K2008-69 × -20826-01-4, K2008-69 × -20826-01-4; Östgöta Brandstodsbolag; Medical Research Council of Southeast Sweden (FORSS); JDRF Wallenberg Foundation, Grant/Award Number: K 98-99D-12813-01A; ALF-and LfoU grants from Region Östergötland and Linköping University, Sweden

## Author contributions

J.T.R performed the data generation, processing, and analysis and prepared the paper and figures; L.F.W.R and M.O. contributed to data analysis and interpretation; J.I. performed the HLA genotyping; M.A.A, D.A.S. and E.W.T contributed to result interpretation; J.L. performed ABIS, designed the study and carried out sample collection, storage and transport; and all authors were involved in paper revisions and approval of the final version.

## Additional information

**Competing interests:** The authors declare no competing interests.

