## [Peer Review File · Nature Communications]

Reviewers' comments:

Reviewer #1, expertise in immune host-microbial interactions (Remarks to the Author):

Overall the study is well explained and executed. This is the first description of an association between HLA and microbiota in a human cohort, making this an important finding. I have only 2 minor critiques

To say all organisms are viable after decades of freezing based on the isolation of 2 strains of Bifidobacteria from one sample is mis-leading. Either remove this to the materials and methods or change the title of this section with a more accurate description "Bifidobacteria can be isolated from a sample..."

The discussion is quite lengthy and there is a lot of it dedicated to celiac disease. Given that this is a study on Type I diabetes a discussion about how these findings could be used to treat or prevent disease would be more appropriate. For instance, are there organisms identified in this study that are lost in high risk T1D individuals that could be used as probiotics? Or are there unique organisms identified from this study that should be targeted.

Reviewer #2, expertise in human microbiome (Remarks to the Author):

Major comments:

It is not plausible that neither antibiotics nor breastfeeding are associated with gut microbiome composition. I propose some additional analyses:

Author report the duration of the breastfeeding did not have effect on the gut microbiome. It is unclear, however, if any of the babies were still breastfed during the sample collection and whether this affected the composition of the stool samples in question.

Rather than correcting for antibiotic use during any time during the first year, I would advice correcting for any recent antibiotic use (e.g. <1 month prior to stool sample collection) which is much more likely to be reflected in the microbial profile of the stool.

According to authors "core microbiome and beta diversity differed with HLA genotype" (Abstract). However, in most statistical tests the authors have stratified HLA genotypes into four different strata based on the HLA conferred risk for T1D. Authors need provide more evidence (data) to support claims that microbial features associate with unique HLA genotypes.

Authors use different measures of beta-diversity in order to test for differences in gut microbial composition between the groups. Authors are correct that the choice of beta-diversity measure can have drastic effect on PERMANOVA results. However, their description of the differences between these measured is inaccurate. Importantly, the robustness of binomial distance to uneven sampling refers to sampling within each ecosystem/community: in macroscopic ecosystems the community composition of each habitat is surveyed by collecting multiple observations and by counting different species (referred to as sampling, sample size etc.). This "sample size" corresponds to the sequencing depth in microbial sequencing study. In this study, there is no sampling bias (but rather the sample size is constant) since the authors have rarified all samples to 10,000 reads. See e.g. [Wolda, H. (1981). Similarity indices, sample size and diversity. *Oecologia* 50, 296–302.] for more complete discussion. Authors' reasoning on the differences between the different beta-diversity metrics thus is not accurate. There are no biases introduced by the different size of the HLA risk class groups. However, the statistical significance obtained is limited by the smaller/smallest group (in pairwise and all-vs-all comparison, respectively). Interpreting the differences between the beta-diversity measures is more complicated than authors acknowledge. To simplify, authors may state, e.g., that Bray-Curtis

dissimilarity is mainly sensitive to highly abundant species, whereas Jaccard Index only measures the overlap of the community members regardless of the relative abundances.

LfSe analysis (line 134-): Please, be more specific when you describe bacterial taxa that are associated with different HLA risk groups. Specifically, some of these taxa have negative (Family XVI, Carboxydocella), other have positive (Romboutsia, Intestinibacter) LDA scores but they are all described to be associated. It would be more informative to describe whether these taxa are increased or decreased in the groups of interest. For "Family XVI", report order or any other named taxonomic rank above, otherwise this remains meaningless.

Prevalence analysis (line 173-): Here, authors use a "new method" (PIME) that is not yet peer-reviewed but submitted to a journal (and not available as a preprint to my knowledge). Based on the materials online in github, it looks to me that this method will effectively do feature selection such that the remaining taxa are those the best separate the groups in question (here, HLA risk classes). It follows that the groups are, by definition, better separated on PCoA plots and by PERMANOVA test. Hence, in this section authors are effectively over-fitting and conclusions pose circular reasoning.

Search for accession code PRJNA10423 in NCBI website returned no hits. The data should be made public for reviewers to check.

Minor comments:

Information provided about the stool samples is incomplete. At what age were the samples collected? N=1 stool sample per individual. Please, elaborate in Methods section.

"Fig. 1 – Alpha diversity is not significant between genetic risk groups": Unclear sentence, suggested edit "No (significant) difference in microbial alpha diversity between genetic risk groups".

UNIVERSITY of FLORIDA
Institute of Food and Agricultural Sciences
Department of Microbiology and Cell Science

1355 Museum Drive
PO Box 110700
Gainesville, FL 32611-0700
352-392-5430
ewt@ufl.edu

April 29, 2019

Nature Communications
One New York Plaza, Suite 4500 New
York, NY 10004-1562

Reviewer 1, expertise in immune host-microbial interactions

- 1) To say all organisms are viable after decades of freezing based on the isolation of 2 strains of Bifidobacteria from one sample is mis-leading. Either remove this to the materials and methods or change the title of this section with a more accurate description “Bifidobacteria can be isolated from a sample...”.**

We have made the change to the appropriate subtitle in the results section concerning the culturing of Bifidobacteria as an indication of sample viability. The subtitle now reads “Bifidobacteria can be isolated from samples stored at -80°C for decades”. Please refer to line 378 in the revised version of the manuscript.

- 2) The discussion is quite lengthy and there is a lot of it dedicated to celiac disease. Given that this is a study on Type I diabetes a discussion about how these findings could be used**

to treat or prevent disease would be more appropriate. For instance, are there organisms identified in this study that are lost in high risk T1D individuals that could be used as probiotics? Or are there unique organisms identified from this study that should be targeted.

We appreciate this feedback, and to decrease the length of the discussion. We have removed one of the paragraphs dedicated to celiac disease. We agree with reviewer 2 that this paragraph was superfluous to the point we are trying to make in the discussion. However, we feel that at least part of the discussion should cover celiac disease, as it is also an autoimmune disease where HLA II is the main genetic risk factor and the microbiome is thought to be an environmental component (much like type 1 diabetes). In addition, ABIS has enrolled participants who are at genetic risk and/or developed celiac autoimmunity, making the analysis presented with this manuscript potentially applicable to other HLA-driven autoimmune disorders aside from type 1 diabetes alone. However, in the current study, which focused only on type 1 diabetes, we were indeed able to identify two bacterial taxa, *Intestinibacter* and *Romboutsia*, that were strongly associated with lower genetic risk for the disease. As a result, we believe these organisms are of great interest as potential probiotics therapies, and this will be one of the avenues pursued as a result of the findings presented in this manuscript.

Reviewer 2, expertise in human microbiome

- 1) It is not plausible that neither antibiotics use nor breastfeeding are associated with gut microbiome composition. I propose some additional analyses:**

Author report the duration of the breastfeeding did not have effect on the gut microbiome. It is unclear, however, if any of the babies were still breastfed during the sample collection and whether this affected the composition of the stool samples in question.

We thank the reviewer for this suggestion. 43 of the 403 samples (10.7%) used in this analysis were collected from infants that were still at least partially breastfed at the time of sample collection (12 months old). Testing using PERMANOVA showed that this variable did have a significant effect on the composition of the gut community. Therefore, this variable is now accounted for and corrected for in all further analyses. In the revised draft, breast feeding is corrected for in both the LefSe and DESeq2 statistical models. Also, any ordination by PCoA is shown both with and without these 43 samples for a side-by-side comparison of the effect of this variable on clustering. Please refer to lines 130-135 for the results of the PERMANOVA.

Rather than correcting for antibiotic use during any time during the first year, I would advice correcting for any recent antibiotic use (e.g. <1 month prior to stool sample collection) which is much more likely to be reflected in the microbial profile of the stool.

As suggested, we tested the effect of antibiotics taken within one month prior to sample collection (>11 months of age, as sample was collected at 12 months from all subjects) on the gut bacterial composition, as this could potentially confound our results. We found that antibiotic use (yes/no) within one month of sample collection did not lead to a significant

difference in composition using PERMANOVA. Please refer to lines 128-130 of the manuscript for these results. Please note that antibiotic use was not associated with future autoimmunity in three type 1 diabetes cohorts, BABYDIET (Endesfelder et al. 2014, Diabetes 63:2006-2014; Endesfelder et al. 2016, Microbiome 4:17), DIPP (Davis-Richardson et al. 2014, Front Microbiol 5:678), and TEDDY (Kemppainen et al. 2017, JAMA Pediatrics 171:1217-1225). All of these are cited in the manuscript. Note that the lack of association with antibiotic use and future autoimmunity does not preclude a role for bacteria in autoimmunity. It suggests that those bacteria that are associated are resistant to the common antibiotics used in pediatrics.

- 2) **According to authors “core microbiome and beta diversity differed with HLA genotype” (Abstract). However, in most statistical tests the authors have stratified HLA genotypes into four different strata based on the HLA conferred risk for T1D. Authors need provide more evidence (data) to support claims that microbial features associate with unique HLA genotypes.**

Because HLA II is highly polymorphic, there are many different genotypes that confer varying levels of genetic risk for type 1 diabetes. This is particularly true for lower risk haplotypes. To simplify the analysis and to provide a sufficient power for the statistical tests used in this work, participants were grouped into four levels of genetic risk. However, we do provide genotype-specific results in the text. Please refer to lines 171-183 for results that show associations specific to certain HLA haplotypes at the ASV level using DESeq2. In this paragraph, we make comparisons between DR3-positive, DR4-positive and DR3/4 genotypes, specifically. Furthermore, the highest-risk group is made up entirely of one genotype (DR3/4). Therefore, all comparisons across these samples, regardless of binning samples in risk groups, are made at the genotypic level.

- 3) **Authors use different measures of beta-diversity in order to test for differences in gut microbial composition between the groups. Authors are correct that the choice of beta-diversity measure can have drastic effect on PERMANOVA results. However, their description of the differences between these measured is inaccurate. Importantly, the robustness of binomial distance to uneven sampling refers to sampling within each ecosystem/community: in macroscopic ecosystems the community composition of each habitat is surveyed by collecting multiple observations and by counting different species (referred to as sampling, sample size etc.). This “sample size” corresponds to the sequencing depth in microbial sequencing study. In this study, there is no sampling bias (but rather the sample size is constant) since the authors have rarified all samples to 10,000 reads. See e.g. [Wolda, H. (1981). Similarity indices, sample size and diversity. Oecologia 50, 296–302.] for more complete discussion. Authors’ reasoning on the differences between the different beta-diversity metrics thus is not accurate. There are no biases introduced by the different size of the HLA risk class groups. However, the statistical significance obtained is limited by the smaller/smallest group (in pairwise and all-vs-all comparison, respectively). Interpreting the differences between the beta-diversity measures is more complicated than authors acknowledge. To simplify, authors may state, e.g., that Bray-Curtis dissimilarity is mainly sensitive to highly abundant**

species, whereas Jaccard Index only measures the overlap of the community members regardless of the relative abundances.

In light of the excellent points and reference provided by reviewer 2, we have modified the language used to describe the similarity indices used in this work. Please refer to the changes in the paragraph within lines 168-174.

- 4) **LEfSe analysis (line 134-): Please, be more specific when you describe bacterial taxa that are associated with different HLA risk groups. Specifically, some of these taxa have negative (Family XVI, Carboxydocella), other have positive (Romboutsia, Intestinibacter) LDA scores but they are all described to be associated. It would be more informative to describe whether these taxa are increased or decreased in the groups of interest. For “Family XVI”, report order or any other named taxonomic rank above, otherwise this remains meaningless.**

In addition to the LDA score results generated by LEfSe, we also incorporated the differential feature plots in the figure to show higher average relative abundance of the taxon with the associated group. Please refer to Fig. 2. “Family XVI” has also been replaced with the next highest rank (order level) “Clostridiales”. Differential feature plots are also made available in the supplementary material for the LEfSe results depicted in Fig. 5.

- 5) **Prevalence analysis (line 173-): Here, authors use a “new method” (PIME) that is not yet peer-reviewed but submitted to a journal (and not available as a preprint to my knowledge). Based on the materials online in github, it looks to me that this method will effectively do feature selection such that the remaining taxa are those the best separate the groups in question (here, HLA risk classes). It follows that the groups are, by definition, better separated on PCoA plots and by PERMANOVA test. Hence, in this section authors are effectively over-fitting and conclusions pose circular reasoning.**

Because the PIME method is not yet published and accessible for review, we decided to remove the analysis done using PIME from the manuscript. This includes the out of bag error calculations generated by random forest and the PERMANOVA applied to prevalence cutoffs. However, we do still provide results based on taking a prevalence filtering approach to the data, which is similar to approaches taken in previous literature (“core microbiome” studies) e.g. (Huse SM, Ye Y, Zhou Y, Fodor AA 2012. A core human microbiome as viewed through 16S rRNA sequence clusters. *PLOS ONE*. 7:e34242). This approach is informative because it highlights those bacterial ASVs that occur often with a particular risk group, which could be important in understanding how HLA II genetic risk impacts the overall presence/absence of members in the gut community. Also, many ASVs are scarcely seen and are less likely to be important to the question of what effect HLA II is having on the microbiome because they are seen so infrequently between individuals with similar HLA backgrounds. We simply want to observe which ASVs appear most often within genetic risk groups and how those ASVs are shared or unique based on genetic risk. We believe this prevalence filtering approach does not pose circular reasoning for it only addresses the question, “How often is this bacterial ASV observed across the population based on genetic risk?”. If prevalence is very low, then this ASV likely isn't very biologically meaningful. If its prevalence is high and across the

population, then the bacterium represented by this ASV is probably worthy of further investigation.

- 6) Search for accession code PRJNA10423 in NCBI website returned no hits. The data should be made public for reviewers to check.**

We apologize as there was a typographical error in the BioProject accession in the original submission. You can find the raw 16S sequence data submitted to SRA under BioProject accession PRJNA510423. The accession has been updated in the manuscript. Please refer to lines 735-736.

- 7) Information provided about the stool samples is incomplete. At what age were the samples collected? N=1 stool sample per individual. Please, elaborate in Methods section.**

We have added additional information about the sample collection to the methods as requested. “All stool samples used in this analysis were collected at 1 year of age and each subject is represented by a single one-year sample.” Please refer to lines 530-532.

- 8) “Fig. 1 – Alpha diversity is not significant between genetic risk groups”: Unclear sentence, suggested edit “No (significant) difference in microbial alpha diversity between genetic risk groups”.**

We have incorporated the reviewer’s suggested edit to the title of Fig. 1. “No significant difference in microbial alpha diversity between genetic risk groups”.

REVIEWERS' COMMENTS:

Reviewer #2 (Remarks to the Author):

Authors have responded to most of my concerns but

important to present

PCoA plots (Figs. 3 & 6): It's important to present PCoA plots such that the unit lengths are equal on both axis (i.e. use `coord_equal()` on `ggplot2` or any equivalent on other plotting tools). Distance between data points is the most essential information presented on PCoA plots and that should be presented faithfully. Now the authors have (most probably without any bad intentions) stretched the y-axis of the PCoA plots in Fig. 3 such that points at the further ends of the y-axis appear to be further away than they actually should be (and this makes the clustering appear more pronounced). To put this another way, if two points are at 5 units distance from each other on y-axis (while x remains equal) they should appear at similar distance to each other as points that are at 5 units distance on x-axis (while y remains equal). Please, correct the scaling of all PCoA plots.

UNIVERSITY of FLORIDA

Institute of Food and Agricultural Sciences

Department of Microbiology and Cell Science

1355 Museum Drive
PO Box 110700
Gainesville, FL 32611-0700
352-392-5430
ewt@ufl.edu

June 17, 2019

Reviewer #2 (Remarks to the Author):

1) Authors have responded to most of my concerns but

important to present

PCoA plots (Figs. 3 & 6): It's important to present PcoA plots such that the unit lengths are equal on both axis (i.e. use coord_equal() on gglot2 or any equivalent on other plotting tools). Distance between data points is the most essential information presented on PcoA plots and that should be presented faithfully. Now the authors have (most probably without any bad intentions) stretched the y-axis of the PcoA plots in Fig. 3 such that points at the further ends of the y-axis appear to be further away than they actually should be (and this makes the clustering appear more pronounced). To put this another way, if two points are at 5 units distance from each other on y-axis (while x remains equal) they should upper at similar distance to each other as points that are at 5 units distance on x-axis (while y remains equal). Please, correct the scaling of all PcoA plots.

We thank reviewer 2 for bringing this error to our attention. To present the data in the most faithful manner possible, the requested changes to each PCoA plot in the manuscript was done using the function provided by reviewer 2 (i.e. coord_equal() in ggplot2). The x and y axes of the PCoA plots are now equal (length ratio = 1).